# Antibody-Based Therapeutics in Breast Cancer: Clinical and Translational Perspectives

**DOI:** 10.3390/antib15010003

**Published:** 2025-12-25

**Authors:** Anna Balata, Katarzyna Pogoda

**Affiliations:** Department of Breast Cancer and Reconstructive Surgery, Maria Sklodowska-Curie National Research Institute of Oncology, 02781 Warsaw, Poland; katarzyna.pogoda@nio.gov.pl

**Keywords:** breast cancer, monoclonal antibodies, bispecific antibodies, antibody–drug conjugates, triple-negative breast cancer, HER2-positive breast cancer, hormone receptor–positive HER2-negative breast cancer

## Abstract

Breast cancer remains the most common malignancy and one of the leading causes of cancer-related death among women worldwide. Advances in antibody-based therapies have improved outcomes across all biological subtypes: HER2-positive, triple-negative, and luminal breast cancer. Monoclonal antibodies such as trastuzumab and pertuzumab have established HER2-targeted therapy as a standard of care, while immune checkpoint inhibitors have introduced immunotherapy into the treatment of triple-negative breast cancer. The emergence of antibody–drug conjugates (ADCs), including trastuzumab deruxtecan, sacituzumab govitecan, and datopotamab deruxtecan, has further expanded the available therapeutic options. Bispecific antibodies represent a new generation of agents with the potential to overcome resistance and enhance immune activation. Despite impressive progress, important challenges remain, including resistance mechanisms and the management of treatment-related toxicities. This review summarizes the biological rationale, clinical evidence, resistance mechanisms, and safety profiles of therapies based on monoclonal antibodies, bispecific antibodies, and antibody–drug conjugates in breast cancer. The development of these treatment modalities fosters the implementation of personalized, immunologically informed treatment strategies that are redefining precision oncology in breast cancer.

## 1. Introduction

Breast cancer is the most commonly diagnosed malignancy in women worldwide and remains one of the leading causes of cancer-related death, with over 2.3 million new cases and nearly 685,000 deaths estimated in 2022 [1]. The disease is heterogeneous and comprises biologically distinct subtypes: hormone receptor–positive/HER2-negative (HR+/HER2−), HER2-positive (HER2+), and triple-negative breast cancer (TNBC). Each subtype is characterized by unique therapeutic vulnerabilities [2].

HER2-targeted antibodies, beginning with trastuzumab and followed by pertuzumab, have established targeted therapy as the standard of care in HER2+ disease [3,4]. In TNBC, immune checkpoint inhibitors have entered clinical practice, initially with atezolizumab in combination with chemotherapy, and currently with pembrolizumab as the preferred agent in both early-stage and advanced disease [4,5]. Antibody–drug conjugates (ADCs), including trastuzumab emtansine (T-DM1), trastuzumab deruxtecan (T-DXd) and sacituzumab govitecan (SG), have further expanded the role of antibody-based therapies. ADCs molecules combine the antigen specificity of monoclonal antibodies with the cytotoxic potency of a chemotherapeutic payload. Each ADC consists of three key components: an antibody recognizing a tumor-associated antigen, a chemical linker that ensures stability in circulation and controlled drug release, and a payload, most often a microtubule inhibitor or topoisomerase I inhibitor, which directly damages tumor cells after internalization. The design of a given ADC determines its efficacy, safety, and its potential to exert a bystander effect, allowing activity in tumors with low or heterogeneous antigen expression [6].

Nevertheless, important challenges remain. Resistance to antibody-based therapy develops through antigen heterogeneity, alterations in downstream signaling pathways, and immune escape mechanisms, while toxicities such as cardiotoxicity, interstitial lung disease (ILD), and cytokine release syndrome (CRS) can significantly limit treatment tolerability. Bispecific antibodies (BsAbs) represent a novel class of agents with the potential to overcome resistance and broaden the therapeutic spectrum by simultaneously targeting two antigens or redirecting immune effector cells [7].

This review provides a clinically oriented and translationally focused overview of ADCs, monoclonal antibodies (mAbs), and BsAbs in breast cancer therapy. We summarize currently available agents, pivotal clinical data, resistance mechanisms, and toxicity profiles, with particular emphasis on their current and emerging roles across molecular subtypes.

## 2. Biological Basis of Antibody Therapy in Breast Cancer

The therapeutic application of antibodies in breast cancer relies on distinct biological mechanisms that link recognition of tumor antigens with effector immune responses or targeted delivery of cytotoxic payloads.

### 2.1. Mechanisms of Action of Monoclonal Antibodies

mAbs exert antitumor effects through several mechanisms. A key mechanism is direct inhibition of oncogenic signaling pathways. Trastuzumab, the first monoclonal antibody introduced into breast cancer treatment, binds to domain IV of the HER2 receptor, while pertuzumab binds to domain II, preventing receptor dimerization and thereby inhibiting the PI3K/AKT and MAPK signaling cascades that drive tumor growth. In addition to direct signaling blockade, antibodies stimulate the immune system. Through their Fc region, they bind Fcγ receptors on effector cells, leading to antibody-dependent cellular cytotoxicity (ADCC) mediated by NK cells and antibody-dependent cellular phagocytosis (ADCP) mediated by macrophages. These immune mechanisms significantly contribute to the overall therapeutic effect of antibody-based treatment by promoting elimination of antibody-coated tumor cells [8,9].

### 2.2. Antibody–Drug Conjugates

As noted above, ADCs combine the antigen specificity of monoclonal antibodies with the cytotoxic potential of chemotherapy, delivered via a specialized chemical payload. A key innovation of newer ADCs, exemplified by T-DXd, is the bystander effect. Membrane-permeable payloads can diffuse into neighboring tumor cells with low or heterogeneous antigen expression, thereby extending efficacy beyond antigen-high populations and allowing activity, for example, in HER2-low breast cancer [10].

### 2.3. Bispecific Antibodies

BsAbs are designed to simultaneously recognize two different epitopes or antigens, which broadens their therapeutic potential. Agents such as zanidatamab bind two non-overlapping HER2 domains, promoting receptor aggregation, internalization, and degradation, ultimately enhancing blockade of oncogenic signaling. Another strategy involves HER2 × HER3 BsAbs, which counter a common resistance mechanism—HER3 overexpression and PI3K/AKT activation—by inhibiting both receptors in parallel. A distinct group of BsAbs is T-cell engagers, such as HER2 × CD3 or TROP-2 × CD3 constructs. These antibodies link a tumor-associated antigen (HER2 or TROP-2) to the CD3 complex on T cells, bringing cytotoxic lymphocytes into close proximity with tumor cells and triggering targeted immune killing. While T-cell–engaging BsAbs are highly promising, their clinical development is limited by the risk of CRS. Ongoing studies are evaluating optimized dosing strategies and stepwise dose escalation to mitigate CRS and improve tolerability [7,11].

### 2.4. Biomarker-Based Patient Selection

Predictive biomarkers are crucial for clinical implementation. HER2 status is determined using IHC and ISH, and recognition of the HER2-low category (IHC 1+ or 2+/ISH−) has expanded indications for ADC therapy [10]. PD-L1 expression is assessed using the Combined Positive Score (CPS). CPS is calculated as the number of PD-L1–positive cells (including tumor cells and immune cells) divided by the total number of viable tumor cells, providing a broader reflection of the tumor microenvironment [8]. Additional biomarkers, such as TILs and ctDNA, are being investigated as tools for refining patient selection and monitoring treatment response.

In summary, antibody-based therapies leverage both direct antitumor mechanisms and immune effector engagement. Understanding these biological principles, together with precise, biomarker-driven patient stratification, is critical for the optimal use of mAbs, ADCs, and BsAbs in breast cancer.

## 3. HER2-Positive and HER2-Low Breast Cancer

### 3.1. Monoclonal Antibodies

Trastuzumab was the first monoclonal antibody to revolutionize the treatment of HER2+ breast cancer. Its initial approval was based on a pivotal trial in metastatic breast cancer with HER2 overexpression conducted by Slamon et al. In this study, adding trastuzumab to standard chemotherapy extended median overall survival (mOS) to 25.1 months compared with 20.3 months for chemotherapy alone (HR 0.80; *p* = 0.046) and doubled the response rate (ORR) (50% vs. 32%) [3]. These results established HER2 as a therapeutic target and trastuzumab as the first targeted therapy in breast cancer.

Trastuzumab was subsequently evaluated in curative settings. In the HERA trial, one and two years of adjuvant trastuzumab were compared with observation. One year of trastuzumab significantly improved disease-free survival (DFS) (HR 0.76, 95% CI 0.68–0.86) and OS (HR 0.74, 95% CI 0.64–0.86) versus observation, whereas extending treatment to two years did not provide additional benefit [12]. Neoadjuvant combination therapy with trastuzumab and pertuzumab was then investigated. In the NeoSphere trial, adding pertuzumab to trastuzumab and docetaxel significantly increased the pathological complete response (pCR) rate, achieving 45.8% compared with 29.0% for trastuzumab plus docetaxel alone (*p* = 0.0141) [13].

In the adjuvant APHINITY trial, one year of trastuzumab-based therapy with or without pertuzumab was compared with long-term follow-up. The addition of pertuzumab resulted in a significant improvement in invasive disease-free survival (iDFS) (10-year iDFS 87.2% vs. 83.8%; HR 0.79), with the greatest benefit observed in node-positive patients [14].

In the metastatic setting, combination strategies were also evaluated. The CLEOPATRA trial confirmed that dual HER2 blockade with pertuzumab and trastuzumab plus docetaxel prolonged mOS to nearly 57 months compared with 40.8 months for trastuzumab plus chemotherapy (HR 0.68; *p* < 0.001), thereby establishing dual HER2 blockade as the standard of care [15].

### 3.2. Antibody–Drug Conjugates

ADCs have become central to the management of HER2+ breast cancer, both in the metastatic and early disease settings. The pivotal EMILIA trial first established the role of T-DM1 in patients with previously treated metastatic disease. In this study, T-DM1 significantly prolonged median progression-free survival (mPFS) to 9.6 months compared with 6.4 months for lapatinib plus capecitabine (HR 0.65; *p* < 0.001) and improved mOS to 30.9 vs. 25.1 months (HR 0.68; *p* < 0.001). The ORR was also higher with T-DM1 (43.6% vs. 30.8%), leading to its adoption as the standard of care in this setting [15].

The benefit of T-DM1 was later extended to earlier disease in the KATHERINE trial. Among patients with residual invasive disease following neoadjuvant trastuzumab-based therapy, adjuvant T-DM1 reduced the risk of recurrence by 46% (HR 0.54; *p* < 0.00001). Seven-year iDFS was 80.8% with T-DM1 vs. 67.1% with trastuzumab. The seven-year OS rates were 89.1% with T-DM1 and 84.4% with trastuzumab (HR 0.66; *p* = 0.0027), establishing T-DM1 as the preferred option for this high-risk group [16].

The development of T-DXd marked a new milestone in HER2+ breast cancer. In the phase III DESTINY-Breast03 trial, T-DXd demonstrated a mPFS of 28.8 months (95% CI 22.4–37.9) vs. 6.8 months (95% CI 5.6–8.2) with T-DM1 (HR = 0.33) [17]. For OS, median values were not reached at the earlier interim cut-off, but a later analysis reported a mOS of 52.6 months (95% CI 48.7-NE) with T-DXd vs. 42.7 months (95% CI 35.4-NE) with T-DM1 (HR = 0.73). The ORR of 79.7% vs. 34.2% [17]. Recently, in the phase III DESTINY-Breast09 trial, T-DXd combined with pertuzumab (T-DXd + P) demonstrated a remarkable improvement in PFS compared with the standard trastuzumab, pertuzumab, taxane (THP) regimen. mPFS reached 40.8 months in the T-Dxd + P arm versus 23.8 months with THP (HR = 0.56; 95% CI 0.46–0.68; *p* < 0.0001). These results redefine the first-line therapeutic landscape for HER2+ metastatic breast cancer, highlighting the potential of ADC-based combinations to surpass traditional dual-antibody chemotherapy [18].

The subsequent DESTINY-Breast04 trial expanded its utility to patients with HER2-low tumors. Here, T-DXd achieved an mPFS of 9.9 vs. 5.1 months (HR 0.50; *p* < 0.001) and improved OS to 23.4 vs. 16.8 months (HR 0.64; *p* = 0.001) [19], establishing HER2-low breast cancer as a therapeutically relevant subgroup and redefining the reach of HER2-targeted therapy.

### 3.3. Bispecific Antibodies

BsAbs represent a distinctly new therapeutic direction. Zanidatamab (KN026), which binds two different HER2 epitopes, enhances receptor aggregation and degradation. In early-phase studies of tumors with HER2 overexpression, ORR ranged from 33–40% [20]. In parallel, T-cell-engaging antibodies such as HER2 × CD3 are under investigation, with preliminary data indicating promising cytotoxic activity [11].

### 3.4. Mechanisms of Resistance and Therapeutic Strategies

Resistance to anti-HER2 therapy develops through multiple mechanisms. Loss or alteration of the target epitope can limit trastuzumab binding. Other mechanisms include activation of compensatory pathways through HER3 and PI3K/AKT, epitope masking by mucins such as MUC4, and intratumoral HER2 heterogeneity [21]. To date, strategies to overcome resistance include dual blockade with trastuzumab and pertuzumab, the use of ADCs with a bystander effect to target HER2-low cells, and combinations with tyrosine kinase inhibitors, exemplified by tucatinib in the HER2CLIMB trial. Tucatinib plus trastuzumab and capecitabine improved mPFS to 7.8 vs. 5.6 months in the control arm (HR 0.54; *p* < 0.001) and prolonged OS, including in patients with brain metastases [22]. HER2 × HER3 BsAbs are also emerging as a promising strategy to counter HER3-driven resistance [23].

### 3.5. Toxicities and Clinical Management

Each class of HER2-targeted agents has a distinct toxicity profile. Trastuzumab is primarily associated with cardiotoxicity. In large adjuvant trials, a ≥10 percentage point decline in LVEF to below 50% was observed in 8–12% of patients, while symptomatic heart failure occurred in 1–2% [14,17]. Pertuzumab is generally well tolerated; the most common adverse event is diarrhea, affecting 65–70% of patients, with grade ≥ 3 diarrhea in 8–10% [15].

T-DM1 is most often associated with thrombocytopenia and hepatotoxicity. In KATHERINE, grade ≥ 3 thrombocytopenia occurred in 5.7%, and elevated liver enzymes in 7–8% of patients [17]. T-DXd carries a notable risk of ILD, reported in 10.5% of patients in DESTINY-Breast03 and 12.1% in DESTINY-Breast04, with grade ≥ 3 events in 1–2% [18,24].

### 3.6. Future Perspectives

Current sequencing strategies in HER2+ breast cancer include trastuzumab, pertuzumab, and a taxane in the first line, followed by T-DXd in the second line and T-DM1 or tucatinib-containing regimens in later lines. In the future, BsAbs such as zanidatamab or T-DXd + pertuzumab–based regimens may replace classical dual blockade. Combinations with immunotherapy, including checkpoint inhibitors, are actively being explored. Increasing emphasis is also placed on molecular profiling and ctDNA-based minimal residual disease monitoring to improve treatment optimization.

## 4. Triple-Negative Breast Cancer

### 4.1. Monoclonal Antibodies

The introduction of monoclonal antibody–based immunotherapy has substantially reshaped the treatment of TNBC. One of the earliest targeted strategies was the combination of bevacizumab and paclitaxel. In the E2100 trial, adding bevacizumab to paclitaxel significantly prolonged PFS (11.8 vs. 5.9 months; HR 0.60; *p* < 0.001), but did not improve OS (26.7 vs. 25.2 months; *p* = 0.16) [25]. Similar findings were observed in the AVADO and RIBBON-1 trials, confirming a PFS benefit without OS improvement. Consequently, despite initial FDA approval, bevacizumab was ultimately withdrawn for breast cancer due to a lack of durable OS benefit.

A major breakthrough came from the IMpassion130 trial, which evaluated atezolizumab plus nab-paclitaxel as first-line treatment for metastatic TNBC. In patients with PD-L1–positive tumors. This combination improved mPFS to 7.5 vs. 5.0 months with chemotherapy alone (HR 0.62; *p* < 0.002) and extended OS to 25.0 vs. 18.0 months (HR 0.71), leading to the incorporation of atezolizumab as a therapeutic option in this setting [26].

The benefit of checkpoint inhibition was further confirmed in the KEYNOTE-355 trial. Pembrolizumab plus chemotherapy improved outcomes in patients with PD-L1 expression defined as CPS ≥ 10. mPFS was 9.7 vs. 5.6 months (HR 0.65; *p* < 0.001), with a favorable OS trend (23.0 vs. 16.1 months; HR 0.73) [27]. These results led to approval of pembrolizumab plus chemotherapy and established this regimen as the standard first-line treatment for PD-L1–positive metastatic TNBC.

The efficacy of immunotherapy was subsequently evaluated in early-stage disease. KEYNOTE-522 assessed pembrolizumab added to neoadjuvant chemotherapy followed by adjuvant pembrolizumab. The addition of immunotherapy significantly increased the pCR rate (64.8% vs. 51.2%) and produced a sustained improvement in event-free survival (EFS), with 5-year EFS of 81.2% vs. 72.2% (HR 0.65; 95% CI 0.51–0.83; *p* = 0.00031). The most recent analysis also demonstrated improved OS, with 5-year OS of 86.6% vs. 81.7% (HR 0.66; 95% CI 0.50–0.87) [28]. Based on these data, pembrolizumab has become part of the standard treatment for patients with high-risk early-stage TNBC.

### 4.2. Antibody–Drug Conjugates

SG was the first ADC approved for advanced TNBC. It comprises an anti–TROP-2 antibody conjugated to SN-38, the active metabolite of irinotecan and a potent topoisomerase I inhibitor. Its efficacy was demonstrated in the ASCENT trial. In heavily pretreated patients, SG significantly improved mPFS (5.6 vs. 1.7 months; HR 0.41; *p* < 0.001) and mOS (12.1 vs. 6.7 months; HR 0.48; *p* < 0.001), with an ORR of 35% vs. 5% compared with a treatment of the physician’s choice [29].

SG was then evaluated in first-line TNBC. In ASCENT-03, among 558 patients ineligible for immunotherapy, SG significantly prolonged PFS compared with chemotherapy (mPFS 9.7 months [95% CI 8.1–11.1] vs. 6.9 months [95% CI 5.6–8.2]; HR 0.62; 95% CI 0.50–0.77; *p* < 0.001). ORR was 48% vs. 46%, with a median duration of response (DoR) of 12.2 vs. 7.2 months, respectively [30]. In ASCENT-04 (KEYNOTE-D19), the combination of SG and pembrolizumab prolonged mPFS by 3.4 months (HR 0.65; 95% CI 0.51–0.84; *p* < 0.001), with a favorable OS trend [31].

Datopotamab deruxtecan (Dato-DXd) is an anti–TROP-2 ADC carrying a topoisomerase I inhibitor payload derived from exatecan. Its mechanism is similar to T-DXd, enabling potent DNA damage via topoisomerase I inhibition. In early-phase studies (TROPION-PanTumor01), Dato-DXd demonstrated activity in heavily pretreated breast cancer, including TNBC, with ORR of 25–34% [32].

In the phase III TROPION-Breast02 trial, in patients with metastatic TNBC ineligible for immunotherapy, Dato-DXd versus investigator’s choice chemotherapy improved mPFS by 5.2 months (10.8 months [95% CI 8.6–13.0] vs. 5.6 months [95% CI 5.0–7.0]; HR 0.57; 95% CI 0.47–0.69; *p* < 0.0001) and mOS to 23.7 months (95% CI 19.8–25.6) vs. 18.7 months (95% CI 16.0–21.8; HR 0.79; 95% CI 0.64–0.98; *p* = 0.0291). These results underscore the therapeutic potential of TROP-2–directed ADCs in the first-line treatment of metastatic TNBC [33].

Even more impressive results were reported in the BEGONIA trial (Arm 7), where Dato-DXd was combined with durvalumab as first-line therapy in metastatic TNBC. Among 62 patients, 87% of whom had low or negative PD-L1 expression, the confirmed ORR reached 79% (95% CI 67–88%), including 6 complete responses. mPFS was 13.8 months (95% CI 11–NC), and median DoR was 15.5 months (95% CI 9.9–NC) [34].

Sacituzumab tirumotecan (sac-TMT) is a novel ADC consisting of an anti–TROP-2 antibody and a topoisomerase I inhibitor. In the phase III OptiTROP-Breast01 trial, in previously treated metastatic TNBC, sac-TMT significantly prolonged PFS (median 6.7 vs. 2.5 months; HR 0.32; 95% CI 0.24–0.44; *p* < 0.00001) and OS (not reached vs. 9.4 months; HR 0.53; 95% CI 0.36–0.78; *p* = 0.0005) compared with investigator’s choice chemotherapy. ORR was 45% vs. 12%, with median DoR of 7.1 vs. 3.0 months [35].

Encouraging results were also observed in the first-line setting. In the phase II OptiTROP-Breast05 trial evaluating sac-TMT as first-line treatment for advanced or metastatic TNBC, ORR reached 71%, with mPFS of 13.4 months. These data suggest that sac-TMT may become a next-generation TROP-2–targeted ADC with activity across multiple lines of therapy in TNBC [36].

HER3 expression is common in breast cancer and is associated with poor prognosis. Patritumab deruxtecan (HER3-DXd) is an investigational HER3-targeted ADC carrying a deruxtecan payload. In the phase I/II U31402-A-J101 study, 182 patients with HER3-expressing metastatic breast cancer received HER3-DXd (1.6–8.0 mg/kg every 3 weeks) after several prior lines of therapy. In the TNBC cohort, ORR was 22.6% with mPFS of 5.5 months [37].

Another investigational ADC is ladiratuzumab vedotin, which targets LIV-1, a zinc transporter highly expressed in breast cancer, conjugated to monomethyl auristatin E (MMAE). MMAE is a synthetic antimitotic agent that inhibits microtubule polymerization, leading to cell cycle arrest and apoptosis. In phase I trials, ladiratuzumab vedotin showed promising activity in metastatic TNBC, with ORR of approximately 28–34% [38].

### 4.3. Bispecific Antibodies

As discussed in HER2+ disease, BsAbs represent an emerging therapeutic strategy. One of the most actively explored approaches in TNBC is T-cell–engaging BsAbs, which link a tumor-associated antigen to the CD3 complex on T cells. This strategy has been successful in hematologic malignancies and is now being evaluated in solid tumors, including TNBC, where TROP-2 × CD3 constructs are of particular interest [39]. In addition to T-cell engagers, BsAbs designed to enhance checkpoint inhibition—such as those targeting PD-1/PD-L1 together with other immunoregulatory pathways—are also under investigation [7].

### 4.4. Mechanisms of Resistance and Strategies

Resistance to immune checkpoint blockade in TNBC is multifactorial. It involves loss of antigen presentation, low tumor mutational burden, and an immunosuppressive tumor microenvironment enriched with regulatory T cells and myeloid-derived suppressor cells [40]. Strategies to overcome resistance include combining checkpoint inhibitors with ADCs, targeted therapies (e.g., PARP inhibitors in BRCA-mutated TNBC), or BsAbs capable of bypassing immune escape mechanisms [41].

### 4.5. Toxicities and Clinical Management

Adverse events associated with checkpoint inhibitors are predominantly immune-mediated. In KEYNOTE-522, grade ≥ 3 adverse events occurred in 77% of patients in the pembrolizumab arm versus 73% in the placebo arm; the most common were neutropenia and anemia, while immune-mediated events included thyroid dysfunction (15%) and pneumonitis (2%) [28].

SG is characterized mainly by hematologic and gastrointestinal toxicity. In ASCENT, grade ≥ 3 neutropenia occurred in 51% of patients, diarrhea in 10%, and febrile neutropenia in 6% [29]. Prophylactic use of granulocyte colony-stimulating factor (G-CSF) and antidiarrheal agents is recommended.

In TROPION-Breast02, Dato-DXd demonstrated a predictable safety profile. The most common adverse events were stomatitis (57%, grade ≥ 3 in 8%) and nausea (45%). Ocular toxicity occurred in approximately 24% of patients, usually mild and reversible, and ILD in 1% [33]. Dental and ocular prophylaxis, as well as early ILD monitoring, are therefore essential.

Despite encouraging results with BsAbs, significant challenges remain. CRS is a major concern with T-cell–redirecting antibodies and resembles immune-related toxicities seen with checkpoint inhibitors. Severe or even fatal immune-mediated events, although rare, require close patient monitoring and timely intervention [42].

### 4.6. Future Perspectives

The integration of immunotherapy and ADCs has already transformed the TNBC treatment landscape. Ongoing trials are exploring neoadjuvant combinations of ADCs with checkpoint inhibitors, as well as BsAbs capable of enhancing immune activation. Future treatment algorithms will likely incorporate pembrolizumab-based regimens, SG, next-generation ADCs, and BsAbs, tailored according to biomarkers such as PD-L1 expression, TILs, and ctDNA.

## 5. Hormone Receptor–Positive/HER2-Negative Breast Cancer

### 5.1. Monoclonal Antibodies

The current standard of care for advanced HR+/HER2− breast cancer is endocrine therapy combined with CDK4/6 inhibitors. Historically, efforts to expand therapeutic options beyond endocrine therapy and chemotherapy included evaluation of monoclonal antibodies targeting angiogenesis or immune pathways. Bevacizumab, previously discussed in TNBC, was also studied in several phase III trials in combination with chemotherapy in HR+ disease. Similar to TNBC, bevacizumab improved PFS (e.g., 10.6 vs. 6.2 months; HR 0.48; *p* < 0.001 in E2100) without an OS benefit, ultimately leading to withdrawal of its approval in this indication [43].

Checkpoint inhibitors have shown more limited efficacy in HR+ disease than in TNBC. In the KEYNOTE-028 trial, which included patients with HR+/HER2− tumors, pembrolizumab monotherapy achieved an ORR of only 12% [44]. More recently, the phase III KEYNOTE-756 trial demonstrated that adding pembrolizumab to neoadjuvant chemotherapy in high-risk, early-stage HR+/HER2− breast cancer significantly increased the pCR rate (24.3% vs. 15.6%), suggesting potential synergy between immunotherapy and chemotherapy in this subtype [45].

### 5.2. Antibody–Drug Conjugates

Despite the positive results of ADCs in advanced HR+/HER2− breast cancer, endocrine therapy plus CDK4/6 inhibition remains the standard first-line approach. Unfortunately, most patients eventually develop resistance, creating a need for additional therapeutic options.

ADCs have shown substantial promise in this setting. The most important agent is T-DXd, which, beyond HER2+ disease, demonstrated significant activity in HER2-low tumors, many of which are HR+/HER2–. In DESTINY-Breast04, T-DXd improved mPFS (10.1 vs. 5.4 months; HR 0.51; *p* < 0.001) and OS (23.9 vs. 17.5 months; HR 0.64; *p* = 0.003) in the HR+/HER2– subgroup [24]. These data established T-DXd as the first targeted therapy for HER2-low breast cancer.

Beyond its activity in TNBC, the role of Dato-DXd has been confirmed in HR+/HER2− disease. In the phase III TROPION-Breast01 trial, Dato-DXd prolonged PFS compared with the investigator’s choice of chemotherapy. mPFS was 6.9 vs. 4.9 months (HR 0.63; 95% CI 0.52–0.76; *p* < 0.0001), and ORR ranged from 26–34% depending on prior CDK4/6 inhibitor exposure. Responses were durable, with a median DoR of 12.7 months. Benefit was consistent across key subgroups, including patients with visceral disease and prior ADC exposure. These findings led to FDA approval of Dato-DXd in 2025 for metastatic HR+/HER2− breast cancer, providing a new option after failure of endocrine therapy and CDK4/6 inhibitors [46].

The efficacy of SG was also evaluated in HR+/HER2− disease. In TROPiCS-02, SG significantly improved PFS compared with investigator’s choice chemotherapy (5.5 vs. 4.0 months; HR 0.66; *p* = 0.0003) and prolonged OS (14.4 vs. 11.2 months; HR 0.79; *p* = 0.02) in patients with endocrine-resistant disease [47].

The phase III OptiTROP-Breast02 trial assessed sac-TMT, previously discussed in TNBC, in patients with HR+/HER2− metastatic breast cancer who had progressed on CDK4/6 inhibitors and chemotherapy. A total of 399 patients were randomized 1:1 to sac-TMT or the investigator’s choice of chemotherapy. Sac-TMT significantly improved PFS (median 8.3 vs. 4.1 months; HR 0.35; 95% CI 0.26–0.48; *p* < 0.0001) and OS (HR 0.33; 95% CI 0.18–0.61) [48].

### 5.3. Bispecific Antibodies

The development of BsAbs in HR+/HER2− breast cancer is still at an early stage. Experimental strategies include TROP-2 × CD3 BsAbs aiming to harness T-cell immunity against TROP-2–positive HR+ tumors and ER × CD3 constructs designed to redirect T cells toward estrogen receptor-driven cancer cells. Preclinical data suggest that these molecules can induce T cell-mediated lysis of ER+ breast cancer cells [39].

Another avenue involves checkpoint-based BsAbs such as PD-1 × CTLA-4 or PD-L1 × 4-1BB, which aim to enhance immunotherapy efficacy in biomarker-selected HR+ subgroups [49].

### 5.4. Mechanisms of Resistance

Resistance in HR+/HER2− breast cancer is multifactorial. Tumors frequently develop endocrine resistance through ESR1 mutations, activation of the PI3K/AKT/mTOR pathway, or cyclin D amplification. These alterations can blunt the efficacy of antibody-based therapies by generating redundant survival pathways or creating an immunosuppressive microenvironment [39]. Strategies to overcome resistance include combining ADCs with endocrine therapy, integrating PI3K or AKT inhibitors with immunotherapy, and sequential use of novel ADCs in TROP-2– or HER3-positive tumors.

### 5.5. Toxicities and Clinical Management

Toxicity profiles of antibody-based therapies in HR+/HER2– breast cancer largely mirror those observed in HER2+ disease and TNBC, given the overlap in agents used.

For T-DXd, ILD remains the most clinically significant toxicity. In DESTINY-Breast04, ILD of any grade occurred in 12.1% of patients, with grade ≥ 3 events in 1.9%. Early recognition of respiratory symptoms and prompt initiation of corticosteroids are crucial in preventing severe ILD. Other frequent adverse events included nausea (73%; grade ≥ 3 in 6%), fatigue (47%; grade ≥ 3 in 5%), and vomiting (44%) [19].

For Dato-DXd, data from TROPION-Breast01 confirm a predictable and manageable toxicity profile consistent with the DXd payload. The most common adverse event was stomatitis (approximately 72% of patients; grade ≥ 3 in 8–10%), followed by nausea (~60%), alopecia (38%), and fatigue (32%) [46]. Most events resolve with dose modification or supportive care, including prophylactic mouth rinses and early use of antiemetics.

In TROPiCS-02 evaluating SG, the most common toxicities were neutropenia (51%; grade ≥ 3 in 49%), diarrhea (62%; grade ≥ 3 in 9%), and nausea (61%), consistent with topoisomerase I inhibition [47]. Prophylactic use of growth factors, adequate hydration, and prompt antidiarrheal therapy are essential.

To date, immune-related adverse events are rare in HR+/HER2− disease due to limited use of immunotherapy, but they may become clinically relevant as trials of checkpoint inhibitors and BsAbs progress in this subtype.

### 5.6. Future Perspectives

The approval of T-DXd in HER2-low disease and Dato-DXd in HR+/HER2− breast cancer heralds a new era of antibody-based therapy in this large patient population. Future strategies will likely include earlier use of ADCs, exploration of sequential ADC approaches, and development of BsAbs targeting immune checkpoints or tumor-associated antigens. A major priority will be identifying predictive biomarkers beyond HER2-low and TROP-2 to enable optimal patient selection.

## 6. Optimizing the Therapeutic Use and Sequencing of Antibodies and Antibody–Drug Conjugates in Breast Cancer

The therapeutic landscape of breast cancer is entering a new era in which monoclonal antibodies and ADCs have become integral components of treatment across all biological subtypes. Despite their high efficacy, the rapid expansion of available agents has outpaced the generation of robust evidence on optimal sequencing, cross-resistance, and patient selection. Current clinical decisions are largely empirical, guided by prior therapies, antigen expression, and toxicity profiles.

An emerging and particularly promising strategy is the combination of ADCs with immunotherapy, which unites targeted cytotoxicity with immune activation. Trials combining ADCs with checkpoint inhibitors, especially in TNBC, have demonstrated encouraging response rates and the potential for durable benefit. Such synergistic approaches may redefine the role of ADCs beyond sequential monotherapy and pave the way for immunogenic, multimodal treatment regimens (Table 1).

As novel targets such as HER3, LIV-1, and claudin 18.2 continue to expand the therapeutic spectrum, the key challenge will be the rational integration and sequencing of these agents to maximize long-term benefit. Ultimately, the evolution of antibody-based therapy in breast cancer reflects a paradigm shift—from static receptor targeting towards dynamic, immune-informed precision oncology.

## Figures and Tables

**Table 1 antibodies-15-00003-t001:** ADCs used in the treatment of breast cancer.

ADC Conjugate	Molecular Target	Mechanism of Action	Dosage	Study Results	Registration
Sacituzumab Govitecan (SG)	TROP-2	Anti-Trop2 antibody linked to SN-38	10 mg/kg i.v. days 1 and 8 of each 21day cycle	Ascent trial: mPFS 5.6 months vs 1.7; TROPIC02 trial: mPFS 5.5 months vs 4	Approved in 2nd line for advanced TNBC and HR+/HER2−
Trastuzumab Deruxtecan (T-DXd)	HER2	Trastuzumab linked to DXd	5.4 mg/kg i.v. every 3 weeks	Destiny-Breast04: mPFS 10 months vs 5.4; Destiny-Breast03: mPFS 28.8 months vs 6.8	Approved for advanced HER2 and HER2-low breast cancer
Trastuzumab Emtansine (T-DM1)	HER2	Trastuzumab linked to emtanzine	3.6 mg/kg i.v. every 3 weeks	EMILIA trial: mPFS 9.6 months vs 6.4 KATHERINE trial: iDFS 80.8% vs 67.1%, OS 89.1% vs 84.4%	Early Her2+; adjuvant for residual disease Metastatic HER2; 2nd and subsequant lines
Datopotomab Deruxtecan (Dato-DXd)	TROP-2	Anti-Trop2 antibody linked to deruxtecan	6 mg/kg i.v. every 3 weeks	Tropion-Breast02: mPFS 10.8 months vs 5.6 Tropion-Breast01: mPFS 6.9 months vs 4.9	HR+/HER2− metastatic breast cancer after prior endocrine-base therapy and chemotherapy
Sacituzumab Tirumotecan (sac-TMT)	TROP-2	Anti-Trop2 antibody linked to tirumotecan	5 mg/kg i.v. every 2 weeks	OptiTROP-Breast 01: mPFS 6.7 months vs 2.5; mOS not reached vs 9.4 months; OptiTROP-Breast05: mPFS 13.4 months	HER2-negative, locally advanced or metastatic breast cancer after receiving prior therapies

## Data Availability

No new data were generated or analyzed in this study. Data sharing is not applicable to this article.

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
