# Peer review of "Antibody-Based Therapeutics in Breast Cancer: Clinical and Translational Perspectives"

_2073-4468, 2025, doi:10.3390/antib15010003_

Round 1

Reviewer 1 Report

Comments and Suggestions for Authors

Balata et al. reviewed antibody–drug conjugates in breast cancer. The review is interesting, well written, and summary the monoclonal and bispecific antibodies for HER2-positive, triple-negative, and hormone receptor–positive/HER2-negative disease. The mechanisms of resistance and safety considerations are also clearly summarized. Below are a few suggestions to further improve the manuscript:

  1. The authors state: “Antibody–drug conjugates, including trastuzumab emtansine (T-DM1), trastuzumab deruxtecan (T-DXd), and sacituzumab govitecan (SG) have further extended the reach of antibody-based therapies, including HER2-low and heavily pretreated subgroups.” It would be helpful to include more detail about the HER2-low population and the heavily pretreated subgroups, as these categories are clinically important.
  2. Table 1 clearly summarizes the ADCs used in breast cancer. It would be interesting to also include a separate table listing bispecific antibodies that have been investigated or approved for breast cancer.
  3. The manuscript would be easier to follow if the authors add numbered section headings throughout the entire text (not only bolded subtitles). For example, assigning a section number to “Future Perspectives” would help guide the reader.

Author Response

Comment 1: The authors state: “Antibody–drug conjugates, including trastuzumab emtansine (T-DM1), trastuzumab deruxtecan (T-DXd), and sacituzumab govitecan (SG) have further extended the reach of antibody-based therapies, including HER2-low and heavily pretreated subgroups.” It would be helpful to include more detail about the HER2-low population and the heavily pretreated subgroups, as these categories are clinically important.

Respond 1: Thank you for this insightful comment. We have expanded the relevant subsections of the manuscript to provide additional detail regarding the clinical relevance of ADCs in HER2-low disease as well as in heavily pretreated patient populations, including clarification of how these groups were defined in key clinical trials. In the Introduction, we also elaborated on the fundamental structure and design principles of ADCs, providing readers with a clearer framework for understanding their therapeutic role. We hope that these revisions address the reviewer’s concerns and will be viewed favorably.

Comment 2: Table 1 clearly summarizes the ADCs used in breast cancer. It would be interesting to also include a separate table listing bispecific antibodies that have been investigated or approved for breast cancer.

Respond 2: Thank you for this thoughtful suggestion. We agree that bispecific antibodies represent an emerging and rapidly evolving therapeutic class in breast cancer. However, given that the majority of BsAbs are still in early-phase clinical development, we felt that a comprehensive table might give a false impression of established clinical applicability.
Instead, we expanded the corresponding section of the manuscript to provide a clearer overview of key BsAb platforms under investigation, including their mechanisms of action and potential clinical relevance. We believe that this narrative approach more accurately reflects the current state of the field, while still addressing the reviewer’s request for additional context on BsAbs.

Comment 3: The manuscript would be easier to follow if the authors add numbered section headings throughout the entire text (not only bolded subtitles). For example, assigning a section number to “Future Perspectives” would help guide the reader.

Respond 3: Thank you for pointing this out. We fully agree that numbered section headings improve readability and structural clarity. Accordingly, we have revised the manuscript to include a consistent numbering system across all main sections and subsections, including the “Future Perspectives” section. We believe this modification will greatly facilitate navigation for the reader.

Reviewer 2 Report

Comments and Suggestions for Authors

The manuscript by Balata et al. provides a comprehensive overview of recent clinical advancements in the development and application of antibody-based therapeutics for breast cancer treatment. It focuses particularly on HER2-positive, triple-negative, and hormone receptor-positive/HER2-negative breast cancer subtypes, summarizing the associated clinical outcomes reported for these therapeutic strategies. The content is thorough, well-structured, and appropriate for the scope of the journal. I believe that the successful publication of this manuscript will offer researchers in the field a clearer perspective and contribute to addressing existing clinical unmet needs. Therefore, I recommend that the editors consider this review manuscript for publication after the authors address the following minor issues:

  1. The title is clear in scope, but its phrasing reads somewhat repetitive for the intended audience.
  2. The manuscript contains several typographical and stylistic inconsistencies, such as missing hyphens, inconsistent subtitle formatting, and incorrect use of abbreviations.
  3. In the Introduction, it would be beneficial to include a brief description of the design and key components of antibody–drug conjugates. This addition would help readers better understand the roles of linkers and payloads/drugs.
  4. Some sections, such as the second paragraph on Page 4, lack appropriate references to substantiate the statements made.
  5. Please ensure that all abbreviations are defined upon their first appearance in the text.
  6. References 15 and 16 appear to be inappropriate, as Reference 15 describes antibody treatment, whereas the cited source pertains specifically to antibody–drug conjugates.

Author Response

Comment 1: The title is clear in scope, but its phrasing reads somewhat repetitive for the intended audience.

Responde 1:  We thank the reviewer for this helpful observation. To reduce redundancy and improve readability, we have revised the title to better reflect the structure of the manuscript while avoiding repetitive phrasing. The new title reads:  Antibody-Based Therapeutics in Breast Cancer: Clinical and Translational Perspectives

Comment 2: The manuscript contains several typographical and stylistic inconsistencies, such as missing hyphens, inconsistent subtitle formatting, and incorrect use of abbreviations.

Responde 2:  We appreciate the reviewer’s careful attention to detail. The manuscript has been thoroughly revised for typographical and stylistic consistency. 

Comment 3: In the Introduction, it would be beneficial to include a brief description of the design and key components of antibody–drug conjugates. This addition would help readers better understand the roles of linkers and payloads/drugs.

Responde 3: We agree with this valuable suggestion. In the revised Introduction, we have added a concise paragraph describing the design and key components of antibody–drug conjugates, including the role of the monoclonal antibody, the chemical linker, and the cytotoxic payload. We also briefly explain how linker stability and payload properties influence efficacy, safety, and the bystander effect. We believe this additional context will help readers better understand the mechanisms and clinical data discussed in the subsequent sections.

Comment 4:  Some sections, such as the second paragraph on Page 4, lack appropriate references to substantiate the statements made.

Reszponde 4: 

We thank the reviewer for this comment. Because the layout and pagination of the manuscript have changed during revision, we were unfortunately not able to identify with certainty which exact passage corresponds to “the second paragraph on Page 4.” In light of this remark, we have therefore re-examined the entire manuscript and carefully checked that all non-trivial clinical and epidemiological statements are supported by appropriate references, and we have slightly adjusted the wording in a few places where the text reflected general background knowledge rather than specific trial data. We hope that this systematic review of the citations addresses the reviewer’s concern, and we would be happy to provide additional, more specific references in a further revision if particular statements are considered insufficiently supported.

Comment 5: Please ensure that all abbreviations are defined upon their first appearance in the text.

Responde 5: We appreciate this important remark. We have systematically reviewed the manuscript to ensure that all abbreviations are defined at their first occurrence in the main text. 

Comment 6: References 15 and 16 appear to be inappropriate, as Reference 15 describes antibody treatment, whereas the cited source pertains specifically to antibody–drug conjugates.

Responde 6: We thank the reviewer for pointing out this inconsistency. We have revised the reference list and corrected the citations so that statements regarding antibody–drug conjugates are now supported by appropriate ADC-specific references, while the antibody-only reference is cited in the relevant context. The numbering of references has been updated accordingly in the revised manuscript.